# Research on the Coastal Marine Environment and Rural Sustainable Development Strategy of Island Countries—Taking the Penghu Islands as an Example

**Chien-Hung Wu**

Department of Marine Recreational, National Penghu University of Science and Technology, Magong 880, Taiwan; wu1023@gms.npu.edu.tw

**Abstract:** In this study, we examined the keelboat industry on Penghu Island in Taiwan as an example to discuss sustainable development strategies for the marine environment and villages. First, three experts were consulted to compile questionnaires. A snowball sampling method was used to collect 278 samples of residents and tourists living in the coastal area. Opinions were collected from 8 residents, crew members, tourists, and scholars. The data were finally summarized and compared by triangulation method and then examined. We found that introducing the keelboat industry could preserve maritime culture, increase local popularity and leisure options for people, create business opportunities, and improve the economy. It could also lead to a loss of coastal architectural features, increased the amount of trash in the community, around the harbor, and on the sea, no improvement in public facilities and medical care, and decreased the willingness of young people to return to their hometowns. Encouraging men to work in tourism-related industries, assisting in balancing job opportunities, strengthening villagers' communication, improving tourists' environmental literacy, adding onboard guides, improving women's professional knowledge of marine ecology and working opportunities for boat maintenance, as well as actively participating in community development planning, can improve the current situation and achieve the goal of sustainable development.

**Keywords:** keelboat; unpowered sailing; sustainable development; green economy

## 1. Introduction

Taiwan is surrounded by sea, with a total coastline of 3086 km along the main island and offshore islands. The climate is between tropical and subtropical, making it very suitable for developing water sports, especially in the Taiwan Strait in the west and the Pacific Ocean in the east. Although Taiwan is committed to promoting leisure and recreational activities at sea, and in 2010, the Development Program for Promoting the Yacht Activities was drafted and passed in anticipation of promoting marine activities in Taiwan as well as promoting the local economy and developing the tourism industry [1], compared to overseas countries, Taiwan has restricted marine and coastline resources for a long time due to policy factors, which delayed developing related tourism resources. Since 2000, the government has been promoting the "National Ocean Policy White Paper" with the concept of establishing a nation by the ocean. In 2004, the concept of marine education was included in the MoE Major Policy Objectives for 2005–2008. The National Ocean Policy Guideline and National Ocean Policy White Paper in 2006 and the Ocean Policy White Paper in 2007 were developed sequentially for the cultivation of talents. In 2007, the National Marine Education Policy White Paper was established [2], and Taiwan officially entered the era of marine resources promotion.

Keelboat sailing is an emerging marine activity in Taiwan. According to their licenses, keelboats are divided into two types: powerboats and yachts, and are limited to the coastal waters within 30 nautical miles, inter-islands of outlying islands, harbors, rivers, and lakes [3]. They are originally evolved from a boat with a ballast keel placed in the bottom,

with a mast in the middle to support the canvas for wind propulsion, and a long deck in the hull as the center of gravity of the boat to avoid capsizing [4,5]. It is a marine transportation vehicle featuring the green energy concept, with the values and characteristics of leisure, sports, education, and industrial development. Originating as a leisure activity exclusive to the rich and gentry [6], it has gradually opened up its popularity as a leisure and tourism activity for all people [7–9] with the improvement of living and economic standards and promoting the government and industries [7].

Taiwan has 592 yachts, 7 international commercial ports, 4 yacht harbors, and 20 fishing harbors open to the public [10]. There are 7 organizations promoting yacht or powerboat activities and 15 sailing training units. The Penghu island area is advantageous in terms of geographical location and is eager to develop marine tourism activities. Although a cross-strait keelboat sailing exchange was held in 2008, most keelboat sailing activities have been mainly independent activities of boat owners, with only one boat used for carrying passengers for business. The annual sailing operations are small, mainly focused on excursions and snorkeling from May to September [11]. Since 2010, the government and the local university of science and technology have invested many funds and resources to hire professional teachers and purchase four keelboats and have trained more than 100 people [8] and achieved many good results in international competitions so far [9]. Local governments and enterprises have also taken advantage of the boom to promote educational and experiential keelboat sailing courses and tourism activities [12,13], which tourists have received well [14]. To date, there is a total demand for 1185 jobs for sailors (55), yacht captains (70), boat builders (80), and shipping managers (980) in 250 shipbuilding and marine transportation-related companies [15]. If unaffected by the COVID-19 epidemic, it is estimated that 220,000 jobs could be created with a value of more than USD 17.24 million [16], which indicates the potential of the sailing industry in Taiwan. Penghu has an inherent geographical advantage and strategic maritime position and was once the trade and shipping hub for both sides of the Taiwan Strait and neighboring countries [17]. According to 2020 statistics, there are approximately 105,000 inhabitants. Its main economic sectors are based on its historical and cultural heritage, its monuments and ancient buildings, its popular crafts and unique winemaking techniques, as well as its rich marine resources [18]. Currently, 73.14% of the population is of working age. In addition to traditional agriculture and fishing, over 70% of the population is engaged in secondary and tertiary industries [19]. The dependence of the local population on the service industry is evident. As a result of this effort, the number of visitors to Penghu also has increased from 1.5 million in 2010 to 2.27 million [20], reaching 1.08 million in the peak season from May to September 2019 [21]. To date, there are 55 hotels and 1023 B&Bs offering 7923 rooms and accommodating 23,197,000 visitors per day [22], with a maximum occupancy rate of 60% during the peak tourist season [22,23]. In addition to the people of Taiwan, it also attracts tourists from mainland China, Hong Kong, Macau, Japan, Europe, North America, and other Asian regions [23].

Tourism activities mainly help to improve the current conditions of the village's economy, community culture, and environmental sanitation, as well as promote local development [24]. Keelboat sailing is a means of transportation with green energy characteristics [4,5] and is gradually becoming a tourist activity that attracts many business opportunities, promoted by governments and private companies [16,25]. However, keelboats need to navigate through vast waters by powered or unpowered means. The navigation process generates oil pollution [26]. The wide and thick hull can easily cause damage to ships or ports [27] and destroy the surrounding waters, marine ecosystems, and village environments [28]. Thus, it can positively and negatively impact the economy, society, village environment, and the surrounding marine environment [27–31].

From an economic perspective, it usually changes employment, wages, consumption, construction, industry, facilities, prices, incentives, hygiene, culture and creativity, leisure activities, community feedback, and policy coordination [32,33]. On the social level, it will change issues, such as visibility, quality of services and activities, policy participation,

tourism organization planning, cultural and architectural characteristics, security mainte-
nance, community building, and public interaction [34–36]. On the environmental level, it
will change the current situation of public transportation, parking and open space, environ-
mental quality of tourists, garbage volume, woodland and ecological habitat, motorcycle
fumes, water and air quality, etc. [34,37,38].

Therefore, the present study considered that the most realistic situation could be
obtained by applying the tourism impact theory to explore developing the coastal area of
Penghu Island from the economic, social, and environmental aspects of keelboat sailing.
The development brings changes to a place and usually takes time to be evidenced. The
changes that occur are usually felt most acutely after the event is over [35,39] and by the
people who live in the area [25,33]. Therefore, the most relevant answer can be obtained
if the impact and effectiveness of local development of the keelboat sailing industry on
the local village and marine environment are examined from the residents' perspective.
However, tourism development aims to create a tourism environment that meets people's
expectations, enhances their satisfaction with the experience, and promotes consumption,
so understanding tourists' perceptions leads to effective decision-making [34]. To achieve
the goal of sustainable development, a consensus between the two parties is required [25,37].
Therefore, it is necessary to analyze the perceptions of residents and tourists and then
identify the consensus between the two parties [30,33] to determine the development goals
of keelboat sailing tourism.

It was also found that the current research on keelboat sailing mainly focused on
psychology [40], disease [41], sports training method [42], maritime medicine [43], and
navigational dynamics [44], with few studies on tourism issues [45] and even fewer on the
topic of keelboat sailing in Penghu. Therefore, we believed that it would be helpful to fill
the research gap by exploring the impact of keelboat sailing on developing villages and the
surrounding environment in Penghu.

## 2. Literature Discussion

### 2.1. The Importance of the Perception of Tourism Development by Different Stakeholders

To sum up the above, water is a tourism resource that is sustainable to use, and
the vast marine resources are even rich in potential [25]. Keelboat sailing is an emerging
marine tourism industry and resource in Taiwan that can provide tourism, transportation,
and educational experiences within a limited scope [12,13]. Hence, using keelboat sailing
resources to develop marine tourism activities will be advantageous and trend in the island
region [29]. Yet, the quality of tourism development needs to be assessed by people's expe-
rience. The effectiveness of local tourism policy development can be indirectly understood
by the post-consumption experience of tourists [24].

However, development brings changes to a place. It usually takes time for the changes
to be evidenced, typically after the activity is over [35,39], and is most profoundly felt by
the people living in the area [33,46]. Therefore, the most relevant answer can be obtained
if the impact and effectiveness of local development of keelboat industry activities on
local villages and the marine environment are examined from residents' perspective. With
the consensus of both sound development, decisions can be established to move towards
sustainable development [25,37].

### 2.2. Economic Impacts of the Village

The economic impact of villages was one of the first issues to be explored [31]. The
main focus was to understand the current situation of tourism development on the eco-
nomic development, commercial trade, and people's income and employment in the local
city [30,47–49].

The impact of economic development can be explored in terms of consumer price,
industrial construction, and community development [33]. It usually changes employment,
wages, consumption, construction, industry, facilities, prices, incentives, hygiene, culture
and creativity, leisure activities, community feedback, and policy coordination [32,33].

### 2.3. Social Impacts of the Village

The social impact of villages is explored in terms of livelihood issues [32]. The main focus is to understand the impact of tourism development on the existing architecture, customs, culture, as well as security of the local community [25,50–52].

The social impact of village development can be analyzed in terms of village building, quality of life, as well as culture and security [30]. It will change issues, such as visibility, quality of services and activities, policy participation, tourism organization planning, cultural and architectural characteristics, security maintenance, community building, and public interaction [34–36].

### 2.4. Environmental Impacts of the Village

The environmental impact of the village is explored in terms of environmental facilities around the community [23]. The main purpose is to analyze the changes in people's living areas and community facilities as a result of tourism development [30,53,54].

The developmental impact of the village environment can be illustrated in terms of sanitation and ecosystems of the village environment [33]. It will change the current situation of public transportation, parking and open space, environmental quality of tourists, garbage volume, woodland and ecological habitat, motorcycle fumes, water and air quality, etc. [30,37,38].

### 2.5. Impact on the Surrounding Marine Ecosystem

Keelboats are a type of marine transportation vehicle with green energy characteristics [4,5,55], which travel between ports or across open seawater by powered or non-powered means [26]. However, the wide and thick hull of these vessels can easily cause damage to vessels or ports [27]. The navigation process can generate oil pollution [26], which can easily have adverse effects on the surrounding natural environment and ecological conditions [28,56–58]. Therefore, the impact of the surrounding marine ecological environment will be investigated in terms of the environmental health and marine ecological environment along with the port.

## 3. Research Methods

### 3.1. Research Process and Framework

This study aimed to analyze the impact of keelboat industry activities on local villages and the marine environment in an island-type country. The literature was consulted [1–16,55] to understand the status of developing the keelboat industry in the local area. The questionnaire instrument was compiled based on [20–58]. Snowball sampling was used to collect questionnaires. SPSS 20.0 statistical software was used to statistically validate the analysis and then combined with expert interviews to provide insights on the statistical results of the questionnaires. After comparing the data with the triangulation validation method [59–61], the impact of keelboat industry activities on the local village and marine environment was investigated. The research architecture is shown in Figure 1.

Based on the above description, the hypotheses were:

H1: Assume that the public believes that there is no difference in the perception of the impact of the heavy sailing industry on the economic development of the village.

H2: Assume that the public believes that there is no difference in the perception of the impact of the heavy sailing industry on the social development of the village.

H3: Assume that the public believes that there is no difference in the perception of the impact of the heavy sailing industry on the development of the village environment.

H4: It is assumed that the public believes that there is no difference in the perception of the impact of the heavy sailing industry on the development of the surrounding coast and the ecological environment.

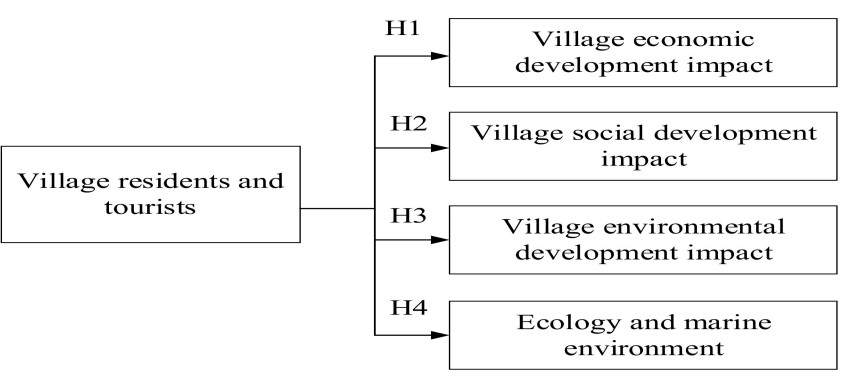

**Figure 1.** Research framework.

*3.2. Research Tools*

The case study was conducted in Penghu, with local villagers, tourists, heavy sailing club members, boat owners, crew, and business owners as the target population, to investigate the impact of keelboat sailing industry development on the coastal villages and ecological environment. A total of 278 questionnaires were distributed between January and February 2021. Samples were obtained from villagers, heavy sailing club members, boat owners, crew, and business owners using a combination of intentional sampling and snowball sampling and from tourists using convenience sampling.

This study aimed to analyze the impact of keelboat activities on local villages and the marine environment in an island country. The questionnaire was divided into 3 parts. The first part was the background information, for understanding gender (male, female), age (under 19, 21–29, 30–39, 40–49, 50–59, 60+), length of residence or travel stay (short-term travel stay of 1 day, short-term travel stay of 1–3 days, short-term travel stay of more than 3 days, permanent or long-term residence of less than 1 year, permanent or long-term (1–4 years of residence, 5–8 years of residence, 9–12 years of residence, 13 years or more of residence), and the identity of experiencing keelboat sailing (Tourists: sailing owners, crew: residents), etc.

The impact of village development was further divided into economic (14 questions), social (11 questions), and environmental (7 questions) aspects with reference to [30,31,46–54]. The questionnaire on the impact of port marine ecosystems (8 questions) was edited with reference to [4,5,26,27,55–58], as shown in Table 1 below.

**Table 1.** Initial questionnaire issue preparation.

| Level | Issue Content | Number of Questions |
|---|---|---|
| Background information | Gender, age, pilot (or passenger/spectator) experiential identity | 1–3 |
| Economic impacts of villages | Increase other job opportunities<br>Increasing yachting-related entrepreneurial opportunities<br>Increasing salary income from work<br>Increasing consumer spending costs<br>Increase tourism development and industry<br>Facilitate the integration of local specialty industries<br>Increase leisure opportunities<br>Improve the maintenance quality of public facilities<br>The benefits of tourism development are clearly returned to the community<br>Increase the convenience of public transportation<br>Increase land and housing prices<br>Increase in local health standards<br>Set up development protection policy<br>Develop creative merchandise. | 4–18 |

**Table 1.** *Cont.*

| Level | Issue Content | Number of Questions |
|---|---|---|
| Social impacts of villages | Contribute to increased tourism visibility<br>Improve the quality of local services and activities<br>Increase the number of signboards or indicators for tourism activities<br>Facilitate the construction of additional facilities in villages for yacht recreation<br>Promote the establishment of yacht tourism development organizations<br>Contribute to the return of young people to their hometowns for development<br>Contribute to the preservation of marine culture<br>Contribute to the preservation of coastal style architecture<br>Promote interaction between tourists and residents<br>Increase the burden of police, firefighter, and security personnel<br>Increase the willingness to revisit or purchase property in coastal areas | 19–29 |
| Environmental impacts of villages | Increased littering by tourists<br>Insufficient space for boat mooring<br>Insufficient tourist resting and parking facilities<br>The environmental quality of the village is affected by tourists<br>Increased pollution from heavy oil spills<br>Increase in tourism waste<br>Maintenance of scenic and historical sites | 30–32 |
| Port and marine environment aspects | Increase of waste garbage on the sea surface of ports<br>Increase in oil pollution on the port surface<br>Overdevelopment of the coast or land around the port<br>Sufficient public toilets around the port<br>Destroying marine habitats<br>Increasing heavy oil pollution<br>Increasing tourism waste<br>Threats from invasive species | 33–41 |

Except for the background information, a 5-point Likert-type scale was used for all issues. A score of 1 represents strong disagreement, and 5 represents strong agreement. The questionnaire was first compiled by consulting relevant literature, and then three experts were sought to validate the content. Fifty questionnaires were then distributed in November 2020 and statistically validated by using SPSS 22.0 statistical software. When the Kaiser–Meyer–Olkin (KMO) value was > 0.06, and the p-value in Bartlett's test was less than 0.01 ($p < 0.01$), the scale was suitable for continuous factor analysis [62,63]. Good reliability of the questionnaire was indicated when the coefficient $\alpha$ was greater than 0.60 [60], and the results of the analysis are shown in Table 2.

The economic questions had a KMO value > 0.912, Bartlett's approximate $\chi^2$ value of 1403.307, a degree of freedom (df) of 91, and a significance of 0.000 ($p < 0.001$), making them suitable for factor analysis. The explained variances of the scale were 55.03%, 6.44%, and 3.746%, with a total explained variance of 65.217%. All questions were retained after factor analysis and considering the understanding of the actual state of economic development. The questions were divided into three areas: consumer prices (4 questions), industrial development (4 questions), and community development (6 questions), containing a total of 15 questions. The Cronbach's $\alpha$ for the three scales was 0.936, 0.934, and 0.940, respectively.

**Table 2.** Questionnaire for perception analyses of the impact of promoting the keelboat industry on developing coastal areas and the marine environment.

| Facets | Issues | Cronbach's $\alpha$ |
|---|---|---|
| Economic impacts of villages | Increase other job opportunities<br>Increasing salary income from work<br>Increasing consumer spending costs<br>Increase tourism development and industry<br>Facilitate the integration of local specialty industries<br>Increase leisure opportunities<br>Improve the maintenance quality of public facilities<br>The benefits of tourism development are clearly returned to the community<br>Increase the convenience of public transportation<br>Increase land and housing prices<br>Increase in local health standards<br>Set up development protection policy<br>Develop creative merchandise. | 0.934–0.940 |
| Social impacts of villages | Contribute to increased tourism visibility<br>Improve the quality of local services and activities<br>Increase the number of signboards or indicators for tourism activities<br>Facilitate the construction of additional facilities in villages for yacht recreation<br>Promote the establishment of yacht tourism development organizations<br>Contribute to the return of young people to their hometowns for development<br>Contribute to the preservation of marine culture<br>Contribute to the preservation of coastal style architecture<br>Promote interaction between tourists and residents<br>Increase the burden of police, firefighter, and security personnel<br>Increase the willingness to revisit or purchase property in coastal areas | 0.935–0.945 |
| Environmental impacts of villages | Increased littering by tourists<br>Insufficient space for boat mooring<br>Insufficient tourist resting and parking facilities<br>The environmental quality of the village is affected by tourists<br>Increased pollution from heavy oil spills<br>Increase in tourism waste<br>Maintenance of scenic and historical sites | 0.866–0.899 |
| Environmental hygiene along the port | Increase of waste garbage on the sea surface of ports<br>Increase in oil pollution on the port surface<br>Overdevelopment of the coast or land around the port<br>Sufficient public toilets around the port | 0.922–0.927 |
| Marine ecological environment | Destroying marine habitats<br>Increasing heavy oil pollution<br>Increasing tourism waste<br>Threats from invasive species | |

The social questions had a KMO value > 0.930, Bartlett's approximate $\chi^2$ value of 1176.718, degree of freedom (df) of 55, and significance of 0.000 ($p < 0.001$), making them suitable for factor analysis. The explained variances of the scale were 61.719%, 5.025%, and 4.254%, with a total explained variance of 70.997%. All questions were retained after factor analysis and consideration of the understanding of the actual state of economic

development. The questions were divided into three areas: village building (5 questions), quality of life (3 questions), and cultural security (3 questions), containing a total of 11 questions. The Cronbach's α for the three scales was 0.935, 0.938, and 0.945, respectively.

The environmental questions had a KMO value > 0.828, Bartlett's approximate χ2 value of 550.085, degrees of freedom (df) of 21, and significance of 0.000 ($p < 0.001$), making them suitable for factor analysis. The explainable variances of the scale were 55.927% and 8.218%, and the total explainable variance was 67.144%. All questions were retained after factor analysis and considering the understanding of the actual state of economic development. The questions were divided into village environmental hygiene (4 questions) and village environmental ecology (3 questions), containing 7 questions. The Cronbach's α of the two scales was 0.866 and 0.899, respectively.

The questions on port and marine environment had a KMO value $E > 0.924$, Bartlett's approximate χ2 value of 998.49, degree of freedom (df) of 28, and significance of 0.000 ($p < 0.001$), making them suitable for factor analysis. The explained variances of the scale were 68.014% and 7.598%, and the total explained variance was 75.613%. All these were retained after factor analysis and considering the understanding of the actual state of economic development. The questions were divided into two areas: port coastal environmental hygiene (4 questions) and marine ecological environment (4 questions), containing a total of 7 questions. The Cronbach's α of the two scales was 0.922 and 0.92, respectively.

Next, we defined the scope of sample collection, targeting those who have operated, ridden, or experienced keelboats as well as coastal residents. A snowball sampling method was used in December 2020 to collect 278 valid questionnaires combined with an online questionnaire platform. SPSS for Windows 22.0 statistical package was used to compile and analyze the validity of the questionnaires, and t-test and ANOVA were used to analyze the questionnaire results. Finally, based on the results of the questionnaire analysis, interviews were conducted using a semi-structured design and open-ended interviews with people who have operated keelboats, as well as coastal residents, boaters, businessmen, tourists, and scholars, with the consent of the respondents, to present their opinions on the questionnaire results. The information from the questionnaire was integrated, and the results were analyzed. The paper was constructed following the order of summarization, organization, and analysis [39]. Lastly, a multivariate validation analysis was used to combine information from different research subjects, research theories and methods, and multiple viewpoints to examine multiple data and compare the results of various studies [39,40] to obtain accurate knowledge and meaning and to investigate the impact of keelboat industry development on developing coastal areas and the marine environment (Table 3).

**Table 3.** Respondent's background information and an overview of the interview outline.

| Identity | Gender | Residence Time/Experience | Identity | Gender | Residence Time/Experience |
|---|---|---|---|---|---|
| Elderly | Male | 25 | Tourist | Male | 40 |
| Elderly | Female | 30 | Tourist | Female | 25 |
| Professor | Male | 15 | Crew | Female | 3 |
| Entrepreneur | Male | 20 | Crew | Female | 2 |

| Construct | Issues |
|---|---|
| Impact of tourism development | 1. How has the introduction of the keelboat industry affected the economic, social, and environmental conditions of coastal villages? Please briefly explain the reasons and make suggestions for improvement. <br> 2. How has the introduction of the keelboat industry affected the environmental hygiene and marine ecological environment in coastal ports? Please briefly explain the reasons and make suggestions for improvement. |

*3.3. Methodology Analysis and Limitations*

The purpose of the study was to investigate the impact of promoting keelboat industry activities on local villages and the marine environment using a mixed research method to investigate local residents and people who have operated or ridden keelboats in the surrounding villages and coastal areas of Penghu Island.

Villagers, tourists, sailors with experience in operating keelboats, boat owners, and business owners were sampled for the study. Since tourists are not registered in the area, they cannot be classified as residents if they only stay for a short period to live or study, such as students studying for a short period. By collecting the opinions of visitors, who have lived in the village for a year, the researchers hope to understand the impact of keelboat sailing development on the current state of the local village and marine environment based on the experiences of their brief stay and the changes before and after their stay.

In addition, promoting keelboat sailing activities is limited by the availability of venues, equipment, and funding. It is not yet widely available to the general public. Therefore, the researcher only surveyed subjects of different genders and identities and excluded subjects of other backgrounds from the discussion.

The initial phase of the study began in 2020. Still, due to the vast area of the study site and the limitations of the research team in terms of manpower, material resources, and funding, it was not possible to complete the sample collection immediately. Moreover, the outbreak of COVID-19 in December 2019 has not yet subsided, leading to limitations in sample collection. Although the information was collected in conjunction with an online questionnaire platform, the number of people with experience in operating or riding keelboats was limited, limiting the number of samples collected by the researchers. The shortcomings of this study will be presented at the end of this paper with suggestions for improvement. We expect subsequent researchers to refine them for improvement.

**4. Result Analysis**

*4.1. Background Analysis*

The total number of samples was 278. The analysis showed that the number of male and female respondents did not differ much, but they were mostly male (51.8%) and female (48.2). The majority (80.6%) were aged 20–29 years old, and the least (0.7%) were aged over 50 years old. Most people who operated (or rode/viewed) the boat were cadets, sailors, boat owners/captains/coaches (50.4%), villagers (41%), and tourists (8.6%).

*4.2. Impact of Keelboat Industry Activities on Local Village Development*

Based on the literature [13–25], the impact of economic development was explored, and more in-depth answers were obtained from the perspective of different identities or positions [13,14,25]. Statistical tests, t-tests, and ANOVA tests were used to analyze the awareness of different genders and keelboat sailing experience identity on the current status of village development. Then the survey results were investigated by interviews, and finally, by multivariate verification [38–40].

4.2.1. Impact on Village Economic Development

The analysis of the respondents' gender, age, and keelboat sailing experiential identity on economic development is shown in Table 4. The results show that most of the respondents believed that it would increase the opportunities for yachting-related entrepreneurship (4.12), leisure (4.3), and creative goods (3.96). However, it would not help raise consumer spending (3.87), improve the maintenance quality of public facilities (3.92), and raise the standard of local sanitation (3.55).

**Table 4.** Perception analysis of the impact on village economic development.

| Level | Issue | μ | Gender (Male:Female) | Experiencer Identity (Tourists: Sailing Owner, Crew: Residents) |
|---|---|---|---|---|
| Civil price | Increase other job opportunities | 4.02 | 0.001 * (male > female) | 0.076 |
| | Increasing salary income from work | 4.12 | 0.011 | 0.004 |
| | Increasing consumer spending costs | 4.03 | 0.000 * (female > male) | 0.010 |
| | Increase tourism development and industry | 3.87 | 0.772 | 0.002 |
| Industry construction | Facilitate the integration of local specialty industries | 4.18 | 0.003 | 0.000 * |
| | Increase leisure opportunities | 4.23 | 0.000 * (male > female) | 0.096 |
| | Improve the maintenance quality of public facilities | 4.30 | 0.001 * (female > male) | 0.101 |
| | The benefits of tourism development are clearly returned to the community | 3.92 | 0.000 * (female > male) | 0.000* |
| Community development | Increase the convenience of public transportation | 3.82 | 0.292 | 0.001* |
| | Increase land and housing prices | 3.68 | 0.015 | 0.254 |
| | Increase in local health standards | 3.55 | 0.425 | 0.009 |
| | Set up development protection policy | 3.76 | 0.000 * (female > male) | 0.000 * |
| | Develop creative merchandise. | 3.96 | 0.725 | 0.031 |

* $p < 0.001$.

Furthermore, increasing other job opportunities, facilitating integrating local specialty industries, increasing job salary income, increasing leisure opportunities, improving the quality of public facility maintenance, and developing protection policy settings were significant ($p < 0.01$), and increasing other job opportunities and facilitating integrating local specialty industries were felt more by males while increasing job salary income, increasing leisure opportunities, improving the quality of public facility maintenance, and developing protection policy were felt more by females. The differences in the perceptions of those who had different keelboat experiences were not significant.

4.2.2. Impact on Village Social Development

The gender, age, and keelboat sailing experiential identity of the respondents were analyzed for their perceptions of the impact on social development, as shown in Table 5. The results showed that most people thought it helped enhance tourism awareness (4.23), preserving maritime culture (4.05), promoting interaction between tourists and residents (3.99), and improving the burden on police and fire safety personnel (3.81). Still, it did not help improve the current situation of tourism event signage or indicator planning (4.12), youth development in their hometowns, and preserving coastal-style architecture (3.96).

In addition, the issues of enhancing establishing a yacht tourism development organization, facilitating developing youths returning to their hometowns, preserving coastal-style architecture, facilitating interaction between tourists and residents, and willingness to revisit or purchase coastal properties were significant ($p < 0.01$), and the issues of facilitating establishing a yacht tourism development organization were more pronounced among males. In contrast, the issues of facilitating developing youths returning to their hometowns, preserving coastal-style architecture, and promoting the interaction between tourists and residents, and willingness to revisit or purchase coastal properties were more pronounced among females. The difference in opinion between those with different keelboat sailing experiential identities was not significant.

**Table 5.** Perception analysis of the impact on village social development.

| Level | Issue | μ | Gender (Male:Female) | Experiencer Identity (Tourists: Sailing Owner, Crew: Residents) |
|---|---|---|---|---|
| Village construction | Contribute to increased tourism visibility | 4.23 | 0.011 | 0.005 |
| | Improve the quality of local services and activities | 4.15 | 0.030 | 0.256 |
| | Increase the number of signboards or indicators for tourism activities | 4.12 | 0.083 | 0.000 * |
| | Facilitate the construction of additional facilities in villages for yacht recreation | 4.16 | 0.879 | 0.012 |
| | Promote establishing yacht tourism development organizations | 4.18 | 0.000 * (male > female) | 0.004 |
| Quality of life | Contribute to the return of young people to their hometowns for development | 3.96 | 0.000 * (female > male) | 0.037 |
| | Contribute to the preservation of marine culture | 4.05 | 0.086 | 0.313 |
| | Contribute to the preservation of coastal style architecture | 3.97 | 0.000 * (female > male) | 0.154 |
| Cultural security | Promote interaction between tourists and residents | 3.99 | 0.001 * (female > male) | 0.009 |
| | Increase the burden of police, firefighter, and security personnel | 3.81 | 0.145 | 0.000 * |
| | Increase the willingness to revisit or purchase property in coastal areas | 3.92 | 0.000 * (female > male) | 0.000 * |

* $p < 0.001$.

### 4.2.3. Impact on Village Environment Development

The gender, age, and keelboat sailing experiential identity of the respondents were analyzed for their perceptions of the impact on environmental development, as shown in Table 6. The results showed that most people thought that tourist rest and parking facilities were still adequate (3.68). Still, the keelboat activities did not help to maintain the landscape and historical sites (3.63) and that tourist waste (3.88) and visitors' littering behavior (3.79) had increased.

**Table 6.** Perception analysis of the impact on village environment development.

| Level | Issue | μ | Gender (Male:Female) | Experiencer Identity (Tourists: Sailing Owner, Crew: Residents) |
|---|---|---|---|---|
| Village environmental sanitation | Increased littering by tourists | 3.81 | 0.000 * (female > male) | 0.058 |
| | Insufficient space for boat mooring | 3.79 | 0.002 | 0.006 |
| | Insufficient tourist resting and parking facilities | 3.70 | 0.026 | 0.069 (sailing owner, crew, residents > tourists) |
| | The environmental quality of the village is affected by tourists | 3.79 | 0.001 * (female > male) | 0.192 |
| Village environment ecology | Increased pollution from heavy oil spills | 3.72 | 0.000 * (female > male) | 0.055 |
| | Increase in tourism waste | 3.86 | 0.000 * (female > male) | 0.533 |
| | Maintenance of scenic and historical sites | 3.65 | 0.020 | 0.004 |

* $p < 0.001$.

Furthermore, the issues of increased littering by tourists, village environmental quality affected by tourists, increased heavy oil emission pollution, and increased tourism waste were significant ($p < 0.01$), with women feeling more about increased littering by tourists, village environmental quality affected by tourists, increased heavy oil emission pollution, and increased tourism waste. Boat owners, boat crews, and residents felt more significantly about the lack of tourist resting and parking facilities.

### 4.3. Impact of Keelboat Industry Activities on the Harbor and Marine Environment

4.3.1. Impact on Environmental Hygiene at the Port

The gender, age, and keelboat sailing experiential identity of the respondents were analyzed for their perceptions of the impact of environmental hygiene at the port, as shown in Table 7. Most of the respondents thought that there was an increase in the amount of waste at the port (3.86) and that there were not enough public toilets around the port (3.32).

**Table 7.** Perception analysis of the impact on environmental hygiene at the port.

| Issue | M | Gender (Male:Female) | Experiencer Identity (Tourists: Sailing Owner, Crew: Residents) |
|---|---|---|---|
| Increase of waste garbage on the sea surface of ports | 3.86 | 0.000 * (female > male) | 0.044 |
| Increase in oil pollution on the port surface | 3.74 | 0.000 * (female > male) | 0.071 (sailing owner, crew, residents > tourists) |
| Overdevelopment of the coast or land around the port | 3.68 | 0.000 * (female > male) | 0.255 (sailing owner, crew, residents > tourists) |
| Sufficient public toilets around the port | 3.32 | 0.293 | 0.172 |

\* $p < 0.001$.

In addition, respondents of a different gender or experiential identities felt significant ($p < 0.01$) about the issues of increased oil pollution in the harbor, increased oil pollution in the harbor, and overdevelopment of the coast or land around the harbor, and women felt more about the issues of increased oil pollution in the harbor, increased oil pollution in the harbor, and overdevelopment of the coast or land around the harbor. Local ship owners, ship crews, and residents felt more significant than tourists about issues, such as increased oil pollution on the sea surface of the port, overdevelopment of the coast or land around the port.

4.3.2. Impact on the Marine Environment

The gender, age, and keelboat sailing experiential identity of respondents were analyzed for their perceptions of the impact on the marine ecosystem, as shown in Table 8.

**Table 8.** Perception analysis of the impact on the marine ecological environment.

| Issue | μ | Gender (Male:Female) | Experiencer Identity (Tourists: Sailing Owner, Crew: Residents) |
|---|---|---|---|
| Destroying marine habitats | 3.64 | 0.000 * (female > male) | 0.005 |
| Increasing heavy oil pollution | 3.57 | 0.000 * (female > male) | 0.091 |
| Increasing tourism waste | 3.72 | 0.000 * (female > male) | 0.001 * |
| Threats from invasive species | 3.38 | 0.000 * (female > male) | 0.001 * |

\* $p < 0.001$.

In addition, the issues of destruction of marine habitat, increase in heavy oil pollution, increase in tourism waste, and the threat of foreign species were significant ($p < 0.01$), and women felt more strongly about the issues of destruction of marine habitat, increase in heavy oil pollution, increase in tourism waste, and the threat of foreign species.

*4.4. Discussion*

4.4.1. Impact of Keelboat Industry Activities on Local Village Development

(a)   Economic Aspects

With the values and characteristics of leisure, sports, education, and industrial development, keelboats sailing also has gradually become a national leisure and tourism activity [6], which can create many business and job opportunities [15,16]. To meet the needs of the keelboat industry and activities, the government and related organizations have established ports, planned parking facilities, and organized competitions and events to attract many tourists to visit, promote business opportunities, and increase employment opportunities.

Nevertheless, although the popularity of the place has increased slightly and prices have risen, the amount of tourist garbage brought by tourists has increased dramatically. The environmental quality is poor, resulting in the destruction of the local environment. There is still room for improvement in sanitation quality. Therefore, the development still cannot solve the increasing cost of consumption, maintenance quality of public facilities, and sanitation standards.

Furthermore, the marine scenery, the scarcity of keelboats, and the unique appearance of the boats create a great tourist attraction. Although the development cannot satisfy everyone's needs, the local government and people are actively developing the available resources to grasp the business opportunities. However, the lack of resources on the island, the hot sun, and the heavy maintenance work on the boats mean that men are more likely to be exposed to them. Women are concerned about the quality of the living environment around them, the development and maintenance of ports, indirectly increasing business and leisure opportunities, and improving the quality of existing public facilities. Therefore, they have different views on job opportunities, integration of local specialty industries, salary and income, leisure opportunities, and public facility maintenance and protection policies. At the same time, there is little difference in the perceptions of those with different experiential identities on the issues. Men have higher perceptions on issues, such as job opportunities and integration of local specialty industries. In comparison, women have higher perceptions on issues, such as job salary and income, leisure opportunities, quality of public facility maintenance, and development and protection policy settings.

Therefore, the researchers recommend: that controlling the existing consumption system, improving public facilities, and introducing medical enterprises or increasing medical facilities to address the shortcomings, increasing job opportunities for women in boat maintenance, operation or interpretation, increasing the wages of men in the workforce, and planning related leisure activities and public facilities can improve the current situation of the above problems.

(b)   Social Aspects

Despite the scarcity of resources on the island, the characteristics and value of keelboat sailing can bring abundant economic benefits to the local government and the public to improve the community environment and enhance the quality of life [22,47–49]. Yet, the small area of developable land has led to the demolition of existing building facilities. The lack of indicators for rest facilities outside the port, as well as the long exposure to the sun and heavy maintenance work, resulting in problems, such as poor planning of tourism activity signboards or indicators, ineffective planning of tourism activity signboards or indicators, the desire of young people to return to their hometown and the disappearance of the characteristic architectural style along the coast, and other problems that still need to be improved.

Furthermore, the unique nature of keelboats and the exposure of international events have led to developing related industries, prompting local citizens to actively form private enterprises or industry development associations in hopes of seizing business opportunities. Although the limited number of boats and the heavy maintenance work make it impossible to satisfy all the people, the diverse business opportunities created by tourism activities, mostly in the service industry, and the distinctive coastal architecture that makes it a local attraction, have increased employment options and opportunities for women, who are attentive and have increased the willingness to travel or buy properties. Therefore, men are more interested in establishing a yacht tourism development organization. At the same time, women feel more strongly about youth returning to their hometowns, preserving coastal-style architecture, promoting interaction between tourists and residents, and being willing to revisit or purchase properties in the coastal area. There is little difference in the opinions of people with different experiential identities.

Therefore, the researchers recommend: the problems identified in the above analysis can be improved by differentiating the tasks of boat maintenance and piloting, adding new boat guides, increasing the signage of the surrounding tourist areas, planning to park and resting spaces nearby, encouraging women to engage in community development and men to engage in tourism industries other than keelboat sailing, maintaining the village culture and architectural opportunities, and balancing the distribution of industrial manpower.

(c)    Environmental Aspects

Despite the lack of resources on the island, with the characteristics and the large productive value of keelboat sailing [6], the local government and organizations have integrated marine ecological resources, built parking and resting facilities, constructed a perfect tourist environment, organized events to enhance the reputation [7], planned marine ecological experience activities, and assisted with educational institutions and the public, in the hope of raising tourists' environmental conservation awareness for marine ecology and the unique features of coastal villages and maintaining the local environment [25].

However, the current planning direction of the port is mainly to meet the needs of tourists. The construction of the pier affects the existing buildings in the vicinity. The style of private buildings cannot be integrated into the current development decision. The influx of people has increased the amount of tourist trash. The small number of local trash cans and the lack of storage space has led to the public perception that the development is not helping to preserve the landscape and historical sites and has increased tourist waste and littering by tourists.

Furthermore, developing the harbor brings many people and business opportunities but simultaneously brings a huge amount of tourism waste. The environmental quality of tourists varies, and there are not enough local consumer and public facilities, few trash cans in public areas, and insufficient space for storage. This has led to higher concern among women about increased littering by tourists, increased pollution from the discharge of heavy oil, and increased tourism waste.

Although the local community is committed to developing the keelboat industry, in an attempt to develop the characteristics of keelboats as one of the local tourism features, the parking facilities around the port are far away from the yacht pier, which is not conducive to tourist consumption and experience, so boat owners, crew members, and residents feel that the planning of tourist leisure parking facilities is still inadequate and has yet to be resolved.

Therefore, the researchers recommend: that future development needs to increase the frequency of communication with villagers to obtain a consensus on development, improve the navigation route of vessels, shorten the process, reduce the leakage of oily waste, increase the frequency of environmental promotion for tourists, and improve the problem of arbitrary disposal of waste, which can improve the trouble of related issues.

4.4.2. Impact of Keelboat Industry Activities on the Harbor and Marine Environment

(a)    Port Environmental Sanitation Aspects

Due to its rarity and uniqueness, the development of the keelboat industry and activities has increased the choice for people to participate in tourism activities and has resulted in an influx of people and business opportunities for the villages [6,7]. However, the influx of tourists has resulted in a huge amount of tourism waste produced and generated by the related industries to meet the demand of tourists. In addition, the number of existing public toilets is not enough to meet the demand of tourists. As a result, the public believes that although the development brings business opportunities to the local community, it also increases the amount of waste on the sea surface of the port and that there is still room for improvement as the public toilets around the port are currently inadequately installed and planned.

The development of tourism can lead to business opportunities [6]. It attracts businesses to invest in various tourism facilities and buildings to meet consumer demand. Still, it also changes the infrastructure of existing villages and the living space of the inhabitants while inconveniencing women's lives and tourism activities. In addition, the rapid increase in the number of tourists has led to an increase in the number of sailing boats, thus increasing the frequency of departure or docking of boats and causing pollution. Although the oil on the sea surface is subtle and requires close observation to detect. In addition, the operation of keelboats is tedious and not easy to maintain and requires staff with sufficient physical strength to be competent. As a result, women are more sensitive to issues, such as the increase of oil pollution on the sea surface of the harbor, the increase of oil pollution on the sea surface of the harbor, and the overdevelopment of the coast or land around the harbor.

The development has led to more frequent ship movements and exchanges. The problem of oil pollution on the port surface has become more obvious due to the docking of existing ships and foreign ships [10]. The development of tourism and industry requires a certain amount of space for construction. In response to the needs of development and business operation decisions, the coast and surrounding land will be developed and damaged, changing the existing environment. Therefore, the increase of oil pollution on the sea surface of the port and the overexploitation of the coast and land around the port are more obvious to local ship owners, crew, and residents than to tourists.

Therefore, the researchers recommend: that adding public restroom facilities, increasing the frequency of environmental literacy promotion for tourists, shortening the navigation distance of vessels in fishing ports, making good use of existing facilities or developing land space, and reducing unnecessary waste of space will help improve the feelings of most people, local ship owners, crew members and residents, and female citizens on related issues.

(b)    Marine Ecological Environment Aspects

Although, the uniqueness of keelboats attracts tourists to experience them, bringing crowds and business opportunities. Local businesses create various products for the consumption of tourists to meet their travel needs [6,7,10]. Because of the small number of public garbage cans around the harbor, which cannot hold a large amount of tourism waste, and the difference in environmental literacy of tourists, the public believes that the development increases tourism waste.

The variety of marine ecosystems, coupled with the fact that the activities of keelboats do not stray from the waters around the island, do not provide an environment for the public to release foreign species illegally or attract invasive species, so the public believes that developing the keelboat industry will not increase foreign species and damage the existing ecology.

Operating keelboats is tedious, heavy, and complicated, which may constrain women's willingness to engage in the training of keelboat operating skills. In addition, the diverse types of work around the coast, compared to the heavy sailing industry, are simple, with

a fixed working environment and long hours, and are conducive to observing changes in the surrounding coast and the ecological environment. As a result, women are more sensitive than men to minor changes in marine habitats, increased pollution from heavy oil emissions, increased tourism waste, and threats from foreign species.

Therefore, the researchers recommend: that strengthening environmental literacy promotion and increasing the number of trash cans in public areas will help reduce the problem of tourism waste overflow. Effective control of vessel navigation planning, shortening the offshore distance, reducing the risk of oily waste leakage, increasing opportunities for women to experience the operating principles of keelboats, and improving knowledge of marine ecology education will help improve women's concerns about the destruction of marine habitats, increased pollution from heavy oil emissions, increased tourism waste, and threats from foreign species.

## 5. Conclusions and Recommendations

### 5.1. Conclusions

The study found that developing the keelboat industry has brought positive uplift to the local area of Penghu but has harmed the economy, society, and environment of the surrounding villages and the marine ecology of the harbor. It is recommended to divide the channel according to the characteristics of the vessels, shorten the course, and set up a special maintenance area. Parking spaces, garbage bins, public toilets, and rest areas for tourists should be increased around the harbor. Communication channels between decision-makers and villagers should be improved. The frequency of environmental knowledge dissemination to tourists should be enhanced. Then, local residents should be encouraged to invest in related industries according to the characteristics of men and women. This will help reduce oil pollution caused by boat operations, reduce garbage generated by tourists, balance the manpower needs of various industries, maintain the local tourism culture, ecology and architectural features, as well as achieve the goal of sustainable development of island areas and marine resources.

### 5.2. Recommendations

Based on the results of the analysis, the study recommends the following:

1. At the economic development level

Control the existing consumption system, improve public facilities, introduce medical enterprises or facilities, increase the job opportunities for females in boat maintenance, operation, or interpretation, increase the salary of males, and plan related leisure activities and public facilities.

2. At the social development level

Increase men's willingness to work in the adjacent industries and help maintain village culture and buildings, and increase women's participation in local development associations.

3. At the environmental development level

Enhance the frequency of environmental promotion for tourists, increase the frequency of communication between the government, enterprises, and villagers, obtain a consensus on development, shorten the distance of boats from or to shore, improve the quality of boat maintenance, and reduce the pollution from oil spills.

4. At the port and marine environment level

Provide more public toilet facilities, strengthen environmental literacy promotion for tourists, shorten the navigation distance of vessels in fishing ports, increase the number of garbage bins in public areas, improve marine ecology education knowledge, make good use of existing facilities or developable land space, and reduce unnecessary space wastage.

5. Research suggestions

Due to the limitation of manpower, material resources, and funding, this study only covers Penghu. Therefore, we propose the following suggestions: to increase the sample data and conduct surveys with other ports or marine recreational facilities in Taiwan; to extend the study to other countries or regions; to further improve the research on developing marine waters and the ecological environment around villages and ports with different research methods and theories.

**Funding:** This research received no external funding.

**Institutional Review Board Statement:** All subjects in the study were anonymously labeled and agreed to participate in the survey.

**Informed Consent Statement:** Informed consent was obtained from all subjects involved in the study.

**Data Availability Statement:** No data support.

**Conflicts of Interest:** The authors declare no conflict of interest.

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
