# Peer review of "Research on the Coastal Marine Environment and Rural Sustainable Development Strategy of Island Countries—Taking the Penghu Islands as an Example"

_water, doi:10.3390/w13101434_

Round 1

Reviewer 1 Report

The main contribution of this paper is that the author emphasizes the importance of research in order to be able to evaluate the role of the keelboat industry on the marine environment and on the rural development as well. However, this paper might have potential to make a useful contribution, but it is still not ready for the publication.

Chapter 1. Introduction

- Literature is insufficient and not updated, especially for the field of sustainable development and tourism in rural areas and marine environment as well.

- in chapter 1.1. – sentence “Currently, there are 592 yachts and 7 international commercial ports, 4 yachting ports, and 20 fishing harbours that are open to the public in Taiwan [6].” – the source for this data are from 2020, so it can’t be used world “currently”.  

- from this isn’t clear – how many boats are there all together, whether boats were built before 2010 on islands or not, and if are, what kind of boats? and what was role in economy? how many people were employed in this sector?

- this chapter missing to provide some key information which are essential for the rest of the paper – for example: demographical data, economical data (structure of economy, and the role of tourism in the economy of islands), when tourism started to develop on islands, the number of international tourist arrivals, the number of accommodation, how many people work in tourism industry, what is the main tourist product, seasonality of tourism, the prices..

- no data of explanation for the marine environment..

Chapter 2. Research Methods

in this chapter - there are no clear answers to some questions, as are:

- how the questionaries’ where distributed

- isn’t clear how tourists (especially those who live there less than one year) can evaluate impacts from the keelboat industry on the marine environment and on the rural development as well – this provides problem in the 2.4. subchapter

- table 3 – all tougher were 278 valid questionnaires, but in this table are only 160 of them. Also, isn’t clear what is meant by “tourists - Residence time/years of work experience”

Chapter 4. Discussion

- the theses presented are not based on the data given in the research, analysed on the begging of the paper or in some references.

Chapter 5. Conclusions and Recommendations

- recommendations have no basis in the obtained research results

- limitations of this research are not given

Author Response

Reviewer 1

Chapter 1. Introduction

- Literature is insufficient and not updated, especially for the field of sustainable development and tourism in rural areas and marine environment as well.

Dear reviewer: I have updated or added references.

- in chapter 1.1. – sentence “Currently, there are 592 yachts and 7 international commercial ports, 4 yachting ports, and 20 fishing harbours that are open to the public in Taiwan [6].” – the source for this data are from 2020, so it can’t be used world “currently”.

Dear reviewer: I have corrected the wording.

- from this isn’t clear – how many boats are there all together, whether boats were built before 2010 on islands or not, and if are, what kind of boats? and what was role in economy? how many people were employed in this sector?

Dear reviewer: I have added the literature, on page 2.

- this chapter missing to provide some key information which are essential for the rest of the paper – for example: demographical data, economical data (structure of economy, and the role of tourism in the economy of islands), when tourism started to develop on islands, the number of international tourist arrivals, the number of accommodation, how many people work in tourism industry, what is the main tourist product, seasonality of tourism, the prices..

Dear reviewer: I have added the literature, on page 2.

- no data of explanation for the marine environment..

Dear reviewer: I have added the literature, on page 2 and 3.

Chapter 2. Research Methods

in this chapter - there are no clear answers to some questions, as are:

- how the questionaries’ where distributed

Dear reviewer: I have added the literature, on page 4.

- isn’t clear how tourists (especially those who live there less than one year) can evaluate impacts from the keelboat industry on the marine environment and on the rural development as well – this provides problem in the 2.4. subchapter

Dear reviewer: I have already made a supplementary explanation. See page 8.

- table 3 – all tougher were 278 valid questionnaires, but in this table are only 160 of them. Also, isn’t clear what is meant by “tourists - Residence time/years of work experience”

Dear reviewer: I have made adjustments. As shown in Table 3.

Chapter 4. Discussion

- the theses presented are not based on the data given in the research, analysed on the begging of the paper or in some references.

Dear reviewer: I have made adjustments. As in section 4.4

Chapter 5. Conclusions and Recommendations

- recommendations have no basis in the obtained research results

Dear reviewer: I have made adjustments and added explanations.As in Chapter 5.

- limitations of this research are not given

Dear reviewer: I have made adjustments and added explanations.As in Chapter 5.

Reviewer 2 Report

Dear authors, 

Your article is very interesting and have a high quality. The purpose of this study was to analyze the impact of keelboat industry activities on local villages and the marine environment on island type country. 
My suggestions for improving quality of your paper are as follows:

  • write a literature review chapter in which you will explore what other authors have researched on the topic. This will increase the number of references in the bibliography.
  • increase the number of researched literature.

These are all comments. I think that by including the chapter Literature review and increasing the number of cited references, you would improve the quality of the article.
Best regards,

Author Response

Reviewer 2

  • write a literature review chapter in which you will explore what other authors have researched on the topic. This will increase the number of references in the bibliography.
  • Dear reviewer: I have made adjustments and added explanations.
  •  
  • increase the number of researched literature.
  • Dear reviewer: I have made adjustments and added explanations.
  •  

These are all comments. I think that by including the chapter Literature review and increasing the number of cited references, you would improve the quality of the article.

  • Dear reviewer: I have made adjustments and added explanations.

Reviewer 3 Report

water-1178718

Dear author,

Many thanks for submitting your paper to Water journal. The MS is at this stage not suitable for a publication. My comments in order to improve your MS:

  1. Introduction:
  • Not a scientific introduction at all, it is rather a description of the region and national/local policies
  • In the introduction section in-depth literature review is missing as well as arguments, why you have carried out the research and what is its added value from a scientific perspective. We perceive later on “objectives” of the study, but cannot perceive their scientific value
  • I would suggest to fully rewrite this section
  • You can have another section 2. – Literature review as many papers were publish about this topic – please search within the “web of science”, “sciencedirect”, and “MDPI journals, consider also Sustainability journal
    • This section could be renamed to the case study, but in my opinion it is too descriptive, actually it does not bring the added value to the paper
  • Review of relevant studies are scarce
  • The hypotheses are not clear (Fig. 1), although explained later on, this needs to be better structured, with a logical flow
  • The H1 and H2 need reconsideration as written like this are not actually scientific hypotheses

Methods

Statistical methods are well explained and described. From the text, we can perceive that the results are statistically valid.

Results

Results are clearly described.

For the entire paper I would suggest not to list subtitle after a title with no text in-between (see also p. 12). I suggest writing some text after each title

Conclusions and recommendations

It was nice that author included recommendations based on the research outcomes. However, conclusions required reconsiderations. Please clearly state your contribution to the scientific community, and write a scientific conclusions, where you try to answer your research questions.

References

  • References and citations should be improved
  • References could be improved with an in-depth scientific research, where author could actually carry out the state-of-the-art in the research, identify gaps and challenges and then author could find which “research gaps” will be fulfilled with this research, and its added value
  • Many “grey” references are included, which should not be a part of a scientific research

Author Response

Review 3

  1. Introduction:
  • Not a scientific introduction at all, it is rather a description of the region and national/local policies
  • In the introduction section in-depth literature review is missing as well as arguments, why you have carried out the research and what is its added value from a scientific perspective. We perceive later on “objectives” of the study, but cannot perceive their scientific value
  • •I would suggest to fully rewrite this section
  • Dear reviewer: I have made adjustments and added explanations. As in Chapter 1.
  •  
  • You can have another section 2. – Literature review as many papers were publish about this topic – please search within the “web of science”, “sciencedirect”, and “MDPI journals, consider also Sustainability journal
  • Dear reviewer: I have made adjustments and added explanations. As in Chapter 2.
  •  
    • This section could be renamed to the case study, but in my opinion it is too descriptive, actually it does not bring the added value to the paper
    • Dear reviewer: I have made adjustments and added explanations.
  • Review of relevant studies are scarce
  • Dear reviewer: I have made adjustments and added explanations.
  • The hypotheses are not clear (Fig. 1), although explained later on, this needs to be better structured, with a logical flow
  • Dear reviewer: I have made adjustments and added explanations.
  • The H1 and H2 need reconsideration as written like this are not actually scientific hypotheses
  • Dear reviewer: I have made adjustments and added explanations.

For the entire paper I would suggest not to list subtitle after a title with no text in-between (see also p. 12). I suggest writing some text after each title

Dear reviewer: I have made adjustments and added explanations.

Conclusions and recommendations

It was nice that author included recommendations based on the research outcomes. However, conclusions required reconsiderations. Please clearly state your contribution to the scientific community, and write a scientific conclusions, where you try to answer your research questions.

 Dear reviewer: I have made adjustments and added explanations. As in Chapter 5.

References

  • References and citations should be improved
  •  Dear reviewer: I have made adjustments and added explanations.
  • References could be improved with an in-depth scientific research, where author could actually carry out the state-of-the-art in the research, identify gaps and challenges and then author could find which “research gaps” will be fulfilled with this research, and its added value
  • Dear reviewer: I have supplemented the research limitations and suggestions.
  •  
  • Many “grey” references are included, which should not be a part of a scientific research

 Dear reviewer: I have made adjustments and added explanations.

Round 2

Reviewer 1 Report

The authors made lot of effort but still some questions important for this paper, remained unanswered.

Chapter 1.

  • from this isn’t clear – were boats built before 2010 on islands or not, and if are, what kind of boats?
  • there are given data for sailing but this is not the same as the boat industry - this data are still missing

- this chapter missing to provide some key information which are essential for the rest of the paper – for example: demographical data, economic data (structure of economy, and the role of tourism in the economy of islands), when tourism started to develop on islands, the number of international tourist arrivals, the number of accommodation, how many people work in tourism industry, what is the main tourist product, seasonality of tourism, the prices.. – without this data Chapter 4.4. can’t be relevant

Chapter 3.

isn’t clear how tourists (especially those who live there less than one year) can evaluate impacts from the keelboat industry on the marine environment and on the rural development as well

- Chapter 4.4. - recommendations have no basis in the obtained research results

Author Response

Reviewer 1

Chapter 1.

    • from this isn’t clear – were boats built before 2010 on islands or not, and if are, what kind of boats?
    • Dear reviewer
  • There were heavy sailing vessels in the local area before 2010, but most of them were ship owners' activities, and only a few were used for passenger-carrying operations.

As in the third paragraph of 1. Introduction (red font) supplementary explanation.

    • there are given data for sailing but this is not the same as the boat industry - this data are still missing
    • Dear reviewer
  • There were heavy sailing vessels in the local area before 2010, but most of them were ship owners' activities, and only a few were used for passenger-carrying operations.

As in the third paragraph of 1. Introduction (red font) supplementary explanation.

  •  

- this chapter missing to provide some key information which are essential for the rest of the paper – for example: demographical data, economic data (structure of economy, and the role of tourism in the economy of islands), when tourism started to develop on islands, the number of international tourist arrivals, the number of accommodation, how many people work in tourism industry, what is the main tourist product, seasonality of tourism, the prices.. – without this data Chapter 4.4. can’t be relevant

Dear reviewer

The impact of the heavy sailing team on the local tourism industry such as hotels, homestays or the number of tourists has been supplemented.

Such as in the text 1. Introduction in the third paragraph (blue font) supplementary explanation.

Chapter 3.

isn’t clear how tourists (especially those who live there less than one year) can evaluate impacts from the keelboat industry on the marine environment and on the rural development as well

Dear reviewer

Such as in the text 3.3. Methodology Analysis and Limitations (red font) supplementary description.

- Chapter 4.4. - recommendations have no basis in the obtained research results

Dear reviewer

The impact of the heavy sailing team on the local tourism industry such as hotels, homestays or the number of tourists has been supplemented.

Such as in the text 1. Introduction in the third paragraph (blue font) supplementary explanation.

Reviewer 2 Report

Dear Authors, 

now your article is fine. 

Best regards, 

Author Response

Dear reviewer

Thank you for your assistance.

Reviewer 3 Report

Dear authors, 

the MS has been substantially improved (sections 1 and 2), considering all my comments.

 However, I would suggest to improve the conclusions - this part was rewritten (1st paragraph), but it is mostly repeating the results part. You have to link you conclusions to your research questions, conclude your thoughts and represent significance of the results. Expose also the added value of your study. 

I would suggest to re-write the conclusions. 

Author Response

Dear reviewer

I have reassembled the research results and wrote a new conclusion statement.

As in the text, the supplementary description in 5. Conclusions and Recommendations (red font).

Round 3

Reviewer 1 Report

Still missing some  key information which are essential for the understanding results of the research and the rest of the paper, as are: demographical data, economical data (structure of economy, and the role of tourism in the economy of islands), how many people work in tourism industry, what is the main tourist product, seasonality of tourism, the prices..

Author Response

Dear reviewer

We have supplemented the relevant information.

Such as p.2, the third paragraph, the content in red.

Thank you for your reminder and assistance.

wish you well.